# LOCALIZED GENERATIONS USING DEEP NEURAL NETWORKS FOR DATASETS WITH STRUCTURAL SIMILARITY

## ABSTRACT

Extracting the hidden structure of the external environment is an essential component of intelligent agents and human learning. The real-world datasets that we are interested in are often characterized by the *locality*: the change in the structural relationship between the data points depending on location in observation space. The local learning approach extracts semantic representations for these datasets by training the embedding model from scratch for each local neighborhood, respectively. However, this approach is only limited to use with a simple model, since the complex model, including deep neural networks, requires a massive amount of data and extended training time. In this study, we overcome this trade-off based on the insight that the real-world dataset often shares some *structural similarity* between each neighborhood. We propose to utilize the embedding model for the other local structure as a weak form of supervision. Our proposed model, the Local VAE, generalize the Variational Autoencoder to have the different model parameters for each local subset and train these local parameters by the gradient-based meta-learning. Our experimental results showed that the Local VAE succeeded in learning the semantic representations for the dataset with local structure, including the 3D Shapes Dataset, and generated high-quality images.

## 1 INTRODUCTION

Extracting the hidden structure of the external environment is essential for achieving intelligent agents and modeling human learning (Kemp & Tenenbaum, 2008; Lake et al., 2015; Higgins et al., 2017; Achille et al., 2018; Saxe et al., 2019). Human beings and/or animals can effectively learn internal representations from a few amounts of experiences. Various methods of nonlinear feature extraction (Maaten & Hinton, 2008; McInnes et al., 2018) are recently proposed to model the complex environment. In addition, thanks to the developments of deep generative models (Kingma & Welling, 2013; Rezende et al., 2014; Goodfellow et al., 2014; Rezende & Mohamed, 2015), we can now handle the high-dimensional dataset on many individual problems.

Although recent studies succeeded in modeling the dataset for the specific problems, there are still challenging properties in real-world. The datasets that we are interested in are often characterized by the *locality*: the change in the structural relationship between the data points depending on location in observation space. For instance, a sequence of experiences gradually changes according to multiple aspects, including time, space, and modality; we need to identify each individual during the development of their faces consistently. Besides, the human-made objects often have multiple color options for the same shape or size. Many studies have incorporated locality for dimensionality reduction and representational learning (Kambhatla & Leen, 1997; Roweis & Saul, 2000; Tenenbaum et al., 2000). For example, combining the local learning approach with classical unsupervised learning algorithms such as PCA significantly improves their model capacity. These studies aim to find mappings between the data and the coordinate space under the assumption that the data space is composed of multiple low-dimensional subspaces (Brand, 2003; Vincent & Bengio, 2003). We refer this approach to as local learning. They usually use a linear projection for embedding models and a $\ell^2$ distance in the input space for a neighborhood construction. In general, we can arbitrarily choose the distance for the neighbor graph, and it affects the quality of the embeddings.

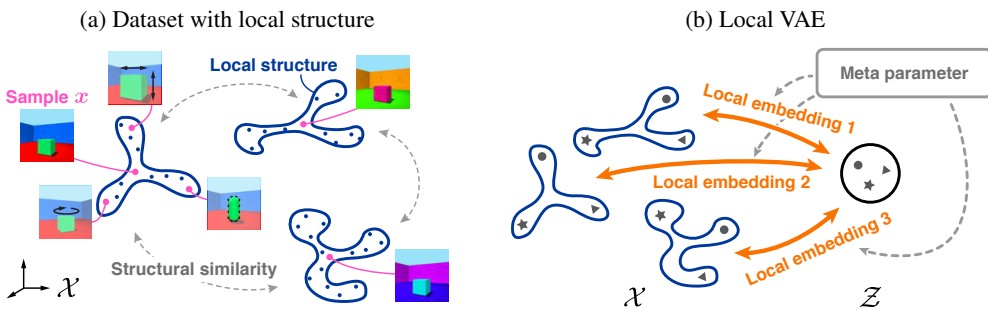

Figure 1: Schematic diagrams of localized generations.

Incorporating the local learning approaches into the training of the deep generative models will give us a new model that has both the capacity for high-dimensional inputs and flexibility for locally changing environments at the same time. However, the integration of these two paradigms is not trivial. The conventional local learning approaches train the different embedding model for each neighborhood from scratch, nevertheless the deep neural networks generally require a large amount of data and take a long training time (LeCun et al., 2015). In other words, local learning approaches learn internal representations for each neighborhood by using only relatively simple models, whereas deep generative models learn one complex representation as a whole with deep neural networks.

To overcome this trade-off, the *structural similarity* between each neighborhood is the key. In the case of the human face, although each face varies greatly depending on age, gender, and etc., there are common facial expressions (Ekman & Keltner, 1997). It is reasonable to expect that each local subspace of the dataset shares some structure since most dataset tends to be governed by the consistent rules of the physical world (Achille et al., 2018). Under the assumptions about the locality and structural similarity, a dataset has two scales of structures: the local structure inside some neighborhood and the global relationship between each neighborhood. Figure 1a visualizes these two scales inside the dataset.

When a dataset has structural similarity, meta-learning is an effective approach. Meta-learning is an algorithm to learn the rules for each task quickly for a dataset consisting of multiple tasks (Schmidhuber, 1987; Bengio et al., 1992; Andrychowicz et al., 2016; Ravi & Larochelle, 2017; Finn et al., 2017). In this study, by considering each local neighborhood as a task for meta-learning, we extract the transferable knowledge between each local structure. We propose to train the meta embedding model, which parameters capture the common local structure and quickly adapt to each subspace by utilizing the structural similarity.

We generalize the typical deep generative model called the Variational Autoencoder (VAE) (Kingma & Welling, 2013; Rezende et al., 2014) to be applicable to the dataset with local structure. We extend the graphical model of the VAE to have different model parameters for each local subset of the dataset (Figure 1b) while keeping to avoid a large amount of computation by using the gradient-based meta-learning (Finn et al., 2017; Grant et al., 2018). By treating the neighborhood of each data point as a task to adopt the meta-learning, we make our proposed Local VAE possible to learn similar structures between neighbors quickly. We evaluate the performance of our proposed model with the 3D Shapes Dataset (Burgess & Kim, 2018) and the concatenated dataset of the Cars3D (Reed et al., 2014) and SmallNORB (LeCun et al., 2004). The numerical experiments shows that the locality enables the model to achieve the disentangled representation for each subspace without any label information.

## 2 BACKGROUND

### 2.1 VARIATIONAL AUTOENCODER

First, we introduce the VAE (Kingma & Welling, 2013; Rezende et al., 2014), which is one of the deep generative models that have been studied extensively in recent years. The objective function

of the VAE is defined as the variational lower bound of the log-likelihood (referred as the evidence lower bound, ELBO) for the dataset. Given dataset $\mathcal{D} = \{\boldsymbol{x}^{(i)}\}_{i=1}^{N}$, ELBO is defined as follows for each $\boldsymbol{x}^{(i)}$;

$$\log p_{\boldsymbol{\theta}}(\boldsymbol{x}^{(i)}) \geq \mathbb{E}_{q_{\boldsymbol{\phi}}(\boldsymbol{z}|\boldsymbol{x}^{(i)})}\Big[\log p_{\boldsymbol{\theta}}(\boldsymbol{x}^{(i)}|\boldsymbol{z})\Big] - D_{\mathrm{KL}}\bigg(q_{\boldsymbol{\phi}}(\boldsymbol{z}|\boldsymbol{x}^{(i)})\Big\|p_{\boldsymbol{\theta}}(\boldsymbol{z})\bigg) = -\mathcal{L}(\boldsymbol{\theta}, \boldsymbol{\phi}; \boldsymbol{x}^{(i)}), \quad (1)$$

where $p_{\boldsymbol{\theta}}(\boldsymbol{x}^{(i)}|\boldsymbol{z})$ is the conditional likelihood referred to as the decoder, and $q_{\boldsymbol{\phi}}(\boldsymbol{z}|\boldsymbol{x}^{(i)})$ is the variational posterior distribution referred to as the encoder. The choice of the prior $p_{\boldsymbol{\theta}}$ is typically the standard normal, and the posterior distribution is also variationally approximated by a Gaussian. This parametric formulation of $q_{\boldsymbol{\phi}}$ is called the reparameterization trick and enables the evaluation of the gradient of the objective function with respect to the network parameters. Overall, we can train the decoder and the encoder networks by taking the minimum of the negative ELBO using the gradient descent method.

## 2.2 MODEL-AGNOSTIC META-LEARNING

Then, to incorporate the local learning approach into the VAE, we utilize the Model-Agnostic Meta-Learning (MAML) (Finn et al., 2017), which is a gradient-based meta-learning algorithm. The goal of MAML is to find task-independent knowledge from a number of previous related tasks. Once the meta-learner learns the task-independent knowledge, it can quickly adapt to a new task using only a few data points and training iterations. For connection to the deep generative models, we introduce the setting based on the maximum likelihood estimation described in Grant et al. (2018) instead of the original MAML formulation. In the setting of MAML, each data point is assumed to be sampled from the task-specific distribution $\boldsymbol{x}^{(i_1)}, \ldots, \boldsymbol{x}^{(i_K)} \sim p_{\mathcal{T}_i}(\boldsymbol{x})$. The MAML objective function in a maximum likelihood setting is

$$\mathcal{L}(\boldsymbol{\theta}) = \frac{1}{N}\sum_{i}\left[\frac{1}{K}\sum_{m} - \log p\Big(\boldsymbol{x}^{(i_{K+m})} \mid \underbrace{\boldsymbol{\theta} - \alpha\boldsymbol{\nabla}_{\boldsymbol{\theta}}\frac{1}{K}\sum_{n} - \log p(\boldsymbol{x}^{(i_n)}|\boldsymbol{\theta})}_{\boldsymbol{\theta}'_{\mathcal{T}_i}}\Big)\right], \qquad (2)$$

where $\boldsymbol{\theta}'_{\mathcal{T}_i}$ is the task-specific parameters after a single batch update by gradient descent from $\boldsymbol{\theta}$. The meta-learner can achieve the parameter $\boldsymbol{\theta}$, which can quickly adapt to new tasks with a small amount of data by optimizing Equation 2 using the gradient method. We note that $\boldsymbol{\theta}$ can be interpreted as the parameters of the prior distribution for the task-specific parameters $\boldsymbol{\theta}_{\mathcal{T}_i}$. By replacing the expectation w.r.t. the original posterior distribution by the maximum likelihood estimate $\int f(\boldsymbol{\theta}_{\mathcal{T}})p(\boldsymbol{\theta}_{\mathcal{T}}|\boldsymbol{\theta})\mathrm{d}\boldsymbol{\theta}_{\mathcal{T}} \simeq f(\boldsymbol{\theta}'_{\mathcal{T}})$, the abovementioned objective function (Equation 2) recovers.

## 3 LOCAL VARIATIONAL AUTOENCODER

In this section, we will present the Local VAE, a variant of the VAE suitable for representation learning of a dataset with local structure.

Here, we extend the objective function of the VAE (Equation 1) to have different parameters for each local subset. We consider the variational lower bound of the log-likelihood for the dataset $\mathcal{D}$, just as with the Vanilla VAE. However, we define the different model parameters $\boldsymbol{\theta}_{N(\boldsymbol{x}^{(i)})}$ and $\boldsymbol{\phi}_{N(\boldsymbol{x}^{(i)})}$ for each neighborhood $N(\boldsymbol{x}^{(i)})$ of the $i$-th data, respectively. Since these parameters are often high-dimensional and require a long time and a large amount of data for training, we give meta parameters $\boldsymbol{\theta}$ and $\boldsymbol{\phi}$ as prior distributions of these local parameters. The overall model performs the probabilistic inference through the conditional distribution from the meta parameters. The variational lower

bound for the log-likelihood can be calculated as follows:

$$\log p_{\boldsymbol{\theta}}(\boldsymbol{x}^{(i)}) = \log \int p(\boldsymbol{x}^{(i)}|\boldsymbol{z}, \boldsymbol{\theta}_{N(\boldsymbol{x}^{(i)})})p(\boldsymbol{z})p(\boldsymbol{\theta}_{N(\boldsymbol{x}^{(i)})}|\boldsymbol{\theta})\mathrm{d}\boldsymbol{z}\mathrm{d}\boldsymbol{\theta}_{N(\boldsymbol{x}^{(i)})} \tag{3}$$

$$\geq \int q(\boldsymbol{z}|\boldsymbol{x}^{(i)}, \boldsymbol{\phi}_{N(\boldsymbol{x}^{(i)})})q(\boldsymbol{\phi}_{N(\boldsymbol{x}^{(i)})}|\boldsymbol{\phi})\times$$

$$\log \frac{p(\boldsymbol{x}^{(i)}|\boldsymbol{z}, \boldsymbol{\theta}_{N(\boldsymbol{x}^{(i)})})p(\boldsymbol{z})p(\boldsymbol{\theta}_{N(\boldsymbol{x}^{(i)})}|\boldsymbol{\theta})}{q(\boldsymbol{z}|\boldsymbol{x}^{(i)}, \boldsymbol{\phi}_{N(\boldsymbol{x}^{(i)})})q(\boldsymbol{\phi}_{N(\boldsymbol{x}^{(i)})}|\boldsymbol{\phi})}\mathrm{d}\boldsymbol{z}\mathrm{d}\boldsymbol{\theta}_{N(\boldsymbol{x}^{(i)})}\mathrm{d}\boldsymbol{\phi}_{N(\boldsymbol{x}^{(i)})} \tag{4}$$

$$= \mathbb{E}_{q(\boldsymbol{z}|\boldsymbol{x}^{(i)}, \boldsymbol{\phi}_{N(\boldsymbol{x}^{(i)})})q(\boldsymbol{\phi}_{N(\boldsymbol{x}^{(i)})}|\boldsymbol{\phi})}\left[\log p(\boldsymbol{x}^{(i)}|\boldsymbol{z}, \boldsymbol{\theta}_{N(\boldsymbol{x}^{(i)})})p(\boldsymbol{\theta}_{N(\boldsymbol{x}^{(i)})}|\boldsymbol{\theta})\right]$$

$$- D_{\mathrm{KL}}\Big(q(\boldsymbol{z}|\boldsymbol{x}^{(i)}, \boldsymbol{\phi}_{N(\boldsymbol{x}^{(i)})})q(\boldsymbol{\phi}_{N(\boldsymbol{x}^{(i)})}|\boldsymbol{\phi})\|p(\boldsymbol{z})\Big), \tag{5}$$

where $p(\boldsymbol{\theta}_{N(\boldsymbol{x}^{(i)})}|\boldsymbol{\theta})$ and $q(\boldsymbol{\phi}_{N(\boldsymbol{x}^{(i)})}|\boldsymbol{\phi})$ are the conditional distribution of the local parameters. We note that the integral variables of the expectation and the Kullback-Leibler divergence in Equation 5 are $\boldsymbol{z}$, $\boldsymbol{\theta}_{N(\boldsymbol{x}^{(i)})}$ and $\boldsymbol{\phi}_{N(\boldsymbol{x}^{(i)})}$.

As we mentioned above, the integral variables of Equation 5 include $\boldsymbol{\theta}_{N(\boldsymbol{x}^{(i)})}$ and $\boldsymbol{\phi}_{N(\boldsymbol{x}^{(i)})}$. This means that Equation 5 needs to take an integral of the model parameters to evaluate the objective function, while the one of Vanilla VAE only requires the Monte Carlo expectation of $\boldsymbol{z}$. Such an integral is unreasonable in deep generative models where model parameters are often high-dimensional. To overcome this problem, we replace this integral with the maximum likelihood estimator updated by the one-step gradient method, as we described in Section 2.2. Let $\mathcal{L}(\boldsymbol{\theta}, \boldsymbol{\phi}; \boldsymbol{x}^{(i)})$ be the negative of the expression obtained by Equation 5. By replacing the integral of $\boldsymbol{\theta}_{N(\boldsymbol{x}^{(i)})}$ and $\boldsymbol{\phi}_{N(\boldsymbol{x}^{(i)})}$ with the maximum likelihood estimator $\boldsymbol{\theta}'_{N(\boldsymbol{x}^{(i)})}$ and $\boldsymbol{\phi}'_{N(\boldsymbol{x}^{(i)})}$, we obtain

$$\mathcal{L}(\boldsymbol{\theta}, \boldsymbol{\phi}; \boldsymbol{x}^{(i)}) \simeq - \mathbb{E}_{q(\boldsymbol{z}|\boldsymbol{x}^{(i)}, \boldsymbol{\phi}'_{N(\boldsymbol{x}^{(i)})})}\left[\log p(\boldsymbol{x}^{(i)}|\boldsymbol{z}, \boldsymbol{\theta}'_{N(\boldsymbol{x}^{(i)})})\right] + D_{\mathrm{KL}}\Big(q(\boldsymbol{z}|\boldsymbol{x}^{(i)}, \boldsymbol{\phi}'_{N(\boldsymbol{x}^{(i)})})\Big\|p_{\boldsymbol{\theta}}(\boldsymbol{z})\Big)$$

$$= \mathcal{L}_g(\boldsymbol{\theta}'_{N(\boldsymbol{x}^{(i)})}, \boldsymbol{\phi}'_{N(\boldsymbol{x}^{(i)})}; \boldsymbol{x}^{(i)}). \tag{6}$$

We note that the integral variable of Equation 6 is now only $\boldsymbol{z}$. The maximum likelihood estimator of the local parameters can be obtained by the following update rule:

$$\boldsymbol{\theta}'_{N(\boldsymbol{x}^{(i)})} = \boldsymbol{\theta} - \alpha\boldsymbol{\nabla}_{\boldsymbol{\theta}}\frac{1}{K}\sum_{\boldsymbol{x} \in N(\boldsymbol{x}^{(i)})}\mathcal{L}(\boldsymbol{\theta}, \boldsymbol{\phi}; \boldsymbol{x}), \tag{7}$$

$$\boldsymbol{\phi}'_{N(\boldsymbol{x}^{(i)})} = \boldsymbol{\phi} - \alpha\boldsymbol{\nabla}_{\boldsymbol{\phi}}\frac{1}{K}\sum_{\boldsymbol{x} \in N(\boldsymbol{x}^{(i)})}\mathcal{L}(\boldsymbol{\theta}, \boldsymbol{\phi}; \boldsymbol{x}), \tag{8}$$

where $K$ is the number of neighborhoods for $\boldsymbol{x}^{(i)}$. $\mathcal{L}(\boldsymbol{\theta}, \boldsymbol{\phi}; \boldsymbol{x})$ in the above equations is the ELBO of Vanilla VAE defined in Equation 1. Algorithm 1 shows the overall algorithm.

From the perspective of the graphical model, our proposed Local VAE algorithm corresponds to the assumption that the dataset approximately lies on multiple subsets and that each subset is generated from different parameters. Alternatively, from the viewpoint of meta-learning, our objective function is consistent with the case of training VAE by MAML when task information is given as a neighbor graph. We can also give the relationship of our model to the conventional local learning approach. Please see Appendix A for the detail.

## 3.1 NEIGHBORHOOD CONSTRUCTION

As we mentioned above, local learning approaches have to construct a neighbor graph before training the model. The conventional approaches often use the $k$-nearest neighbor graph build on the original data space. In general, we can make arbitrarily choice how to construct the neighborhood, and it affects the quality of the embeddings. We evaluated two types of neighborhood in the following experiments: *synthetic neighborhood by sampling* and *k-nearest neighborhood on latent space*. In the synthetic neighborhood by sampling, we sampled $K$ different examples for each $\boldsymbol{x}^{(i)}$ from the noise distribution assumed as the observation process of the data and used these examples as

---

**Algorithm 1** Optimization of Local VAEs

---

1: **while** until converge **do**
2:     **for** $\boldsymbol{x}^{(i)}$ in mini-batch $\mathcal{B}$ **do**
3:         Sample $K$-points from the neighborhood of $\boldsymbol{x}^{(i)}$: $\boldsymbol{x}^{(1)}, \ldots, \boldsymbol{x}^{(K)} \sim N(\boldsymbol{x}^{(i)})$.
4:         Evaluate the local objective $\mathcal{L}(\boldsymbol{\theta}, \boldsymbol{\phi}; \boldsymbol{x})$ for the $K$-neighborhood w.r.t. the meta parameters based on Equation 1.
5:         Update the local parameters:
        $\boldsymbol{\theta}_{N(\boldsymbol{x}^{(i)})} \leftarrow \boldsymbol{\theta} - \alpha \boldsymbol{\nabla}_{\boldsymbol{\theta}} \frac{1}{K} \sum_{\boldsymbol{x} \in N(\boldsymbol{x}^{(i)})} \mathcal{L}(\boldsymbol{\theta}, \boldsymbol{\phi}; \boldsymbol{x})$,
        $\boldsymbol{\phi}_{N(\boldsymbol{x}^{(i)})} \leftarrow \boldsymbol{\phi} - \alpha \boldsymbol{\nabla}_{\boldsymbol{\phi}} \frac{1}{K} \sum_{\boldsymbol{x} \in N(\boldsymbol{x}^{(i)})} \mathcal{L}(\boldsymbol{\theta}, \boldsymbol{\phi}; \boldsymbol{x})$.
6:         Evaluate the global objective $\mathcal{L}_g(\boldsymbol{\theta}_{N(\boldsymbol{x}^{(i)})}, \boldsymbol{\phi}_{N(\boldsymbol{x}^{(i)})}; \boldsymbol{x}^{(i)})$ for $i$-th data w.r.t. the local parameters based on Equation 6.
7:     **end for**
8:     Update the meta parameters:
    $\boldsymbol{\theta} \leftarrow \boldsymbol{\theta} - \eta \boldsymbol{\nabla}_{\boldsymbol{\theta}} \frac{1}{|\mathcal{B}|} \sum_{i \in \mathcal{B}} \mathcal{L}_g(\boldsymbol{\theta}_{N(\boldsymbol{x}^{(i)})}, \boldsymbol{\phi}_{N(\boldsymbol{x}^{(i)})}; \boldsymbol{x}^{(i)})$,
    $\boldsymbol{\phi} \leftarrow \boldsymbol{\phi} - \eta \boldsymbol{\nabla}_{\boldsymbol{\phi}} \frac{1}{|\mathcal{B}|} \sum_{i \in \mathcal{B}} \mathcal{L}_g(\boldsymbol{\theta}_{N(\boldsymbol{x}^{(i)})}, \boldsymbol{\phi}_{N(\boldsymbol{x}^{(i)})}; \boldsymbol{x}^{(i)})$.
9: **end while**

---

the neighborhood of $\boldsymbol{x}^{(i)}$. We considered that this method is effective when the data is densely distributed in the observation space and used this method for 3D Shapes Dataset to omitting the time to construct the neighbor graph for the large dataset. On the other hand, in the $k$-nearest neighborhood on latent space, we used the $k$-nearest neighbor graph builds on the latent space of the VAE. We can expect that we can obtain a neighborhood that follows our intuition by using the distance on latent space rather than the input space (Caron et al., 2019). In the experiment on the CarsNORB Dataset, which we will describe later, we used Faiss (Johnson et al., 2017) for similarity search and continuously updated the latent code for each iteration during the training phase.

## 4 RELATED WORKS

In this study, we employed the gradient-based meta-learning method MAML (Finn et al., 2017) and its probabilistic formulation (Grant et al., 2018) to find local parameters from a few data points. Recently, several studies (Hsu et al., 2019; Metz et al., 2019) proposed the integration of unsupervised learning and meta-learning from another perspective. Hsu et al. (2019) proposed the algorithm for generating MAML task information by utilizing embedded similarity information created with unsupervised learning. In contrast to this case of using unsupervised learning **for** meta-learning, we used meta-learning to perform unsupervised learning. In addition, Metz et al. (2019) proposed a way to seek the objective function itself for representation learning with meta-learning.

The local learning approaches, including LLE (Roweis & Saul, 2000) and Isomap (Tenenbaum et al., 2000), are deeply related to our work. We discuss the relationship between LLE and our work in Appendix A in detail. Besides, the extension of generative models to make them applicable for structured datasets has recently been extensively studied. The generalization of the latent space of the VAE to a non-Euclidean space such as a spherical surface (Davidson et al., 2018), hyperbolic space (Ovinnikov, 2019; Nagano et al., 2019; Mathieu et al., 2019), or discrete space (Jang et al., 2017; Rolfe, 2017) was proposed.

The property of disentanglement has attracted notable attention in structure extraction using VAEs as described above (Higgins et al., 2017; Burgess et al., 2017; Kim & Mnih, 2018; Chen et al., 2018; Kumar et al., 2018; Locatello et al., 2019). Most of the proposed models try to realize disentanglement representation by modifying the penalty term of the objective function or network architectures. On the other hand, our approach focuses on how to learn parameters suitable for disentangled (local) representations so that we can utilize both these aforementioned techniques and our proposed method at the same time.

From the viewpoint of generating data by a deep generative model with some supervision, conditional generation is commonly practiced (Kingma et al., 2014; Sohn et al., 2015; Mirza & Osindero, 2014). Our method is similar to these approaches in that the density function is conditioned on the

| (a) $\alpha = 0$ (Vanilla) | (b) $\alpha = 1\mathrm{e}{-3}$ | (c) $\alpha = 1$ |
|---|---|---|

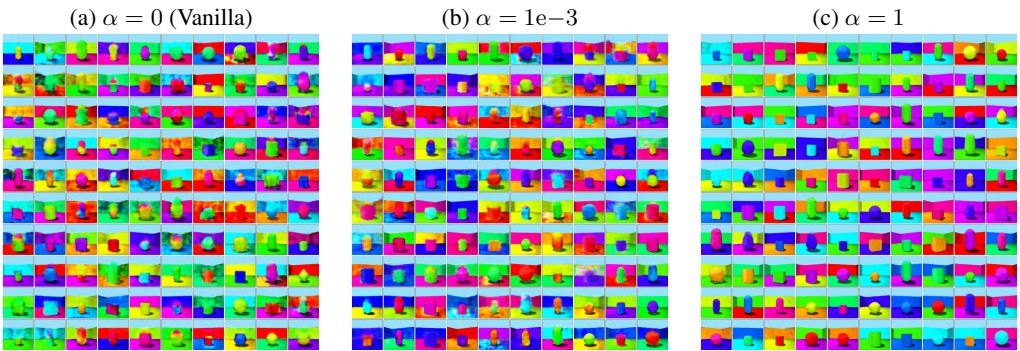

Figure 2: Qualitative evaluation of the randomly-selected conditional prior samples.

neighborhood of the specific data point. While conventional conditional generation generates data by only one model parameter with the known class label as an additional latent code, our proposed model has different network parameters for each neighborhood. Moreover, our approach is more applicable than conventional methods since our approach does not need any class label information.

## 5 NUMERICAL EVALUATIONS

### 5.1 3D SHAPES DATASET

Here, we numerically evaluate the performance of the Local VAE. We use the 3D Shapes Dataset (Burgess & Kim, 2018), which has a clear disentangled property. The disentangled property can be interpreted as the simplest case of the local structure. The disentangled dataset is assumed to be able to control by a small number of factors. Since these factors alter the observation in data space, and the scale of them is different one by one, we can interpret the factors which significantly affect the observation as the global features and other factors as the local features.

We followed all the experimental settings in Locatello et al. (2019), except the batch size and the number of tasks, to eliminate effects outside the proposed method as much as possible. Please see Appendix B for the detail. We qualitatively assess the generated images and quantitatively evaluate the model performance by using the disentanglement metric (DCI scores) proposed by Eastwood & Williams (2018) and the Fréchet Inception Distance (FID) (Heusel et al., 2017).

Figure 2 shows the conditionally generated images of the trained models. At the inference phase, the model obtains the local parameters $\boldsymbol{\theta}_{N(\boldsymbol{x}^{(i)})}$ by applying one-step gradient descent using the randomly selected training data and generates images from these local parameters. We trained multiple models with different values of $\alpha$, which is the hyperparameter of Local VAEs. Note that the original objective function of Vanilla VAEs recovers in the case of $\alpha = 0$ since the local parameters are strictly consistent with the meta parameters. According to the subjective assessment, the quality of generated images is better when $\alpha$ is large.

There could be a concern that overfitting caused the result above. If the model obtains local parameters that perfectly generate only the training sample to be referenced, the quality of the generated image will be superficially high. To exclude this possibility, we visualized the reference samples and their corresponding generated images of the model with $\alpha = 1$ in Figure 3. The leftmost column shows the reference training samples used for the conditional localized generations. Each row visualizes the generated images conditioned on the reference sample in the left column. We randomly picked ten latent codes $\boldsymbol{z}^1, \ldots, \boldsymbol{z}^{10}$ from the prior distribution, and then used these codes for every conditional generation. In other words, the images shown in the same column share their latent code. According to the figure, the trained model generated clearly different images in the same row conditioned on one training sample. This result strongly suggests that the Local VAE model did not overfit to the specific data. Moreover, the shape, angle, and size of the object were the same, and only the color was different in each column. These results suggest that the model trained by the proposed method segregated color information as global features and other information as local

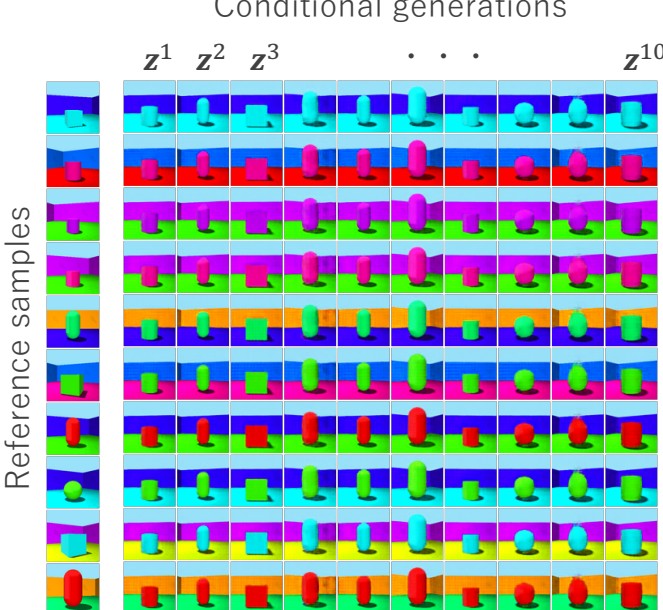

Figure 3: The reference training samples and their corresponding generated images of the Local VAE with $\alpha =1$. The leftmost column shows the reference training samples. Each row visualizes the generated images conditioned on the reference sample. The images shown in the same column share their latent code. The trained model extracted the color information as the global features and other information as the local features.

features, and obtained an internal representation independent of the global features. Note that Local VAEs only use neighborhood relationships and **do not use any label information**.

Then, we qualitatively evaluated the latent representations of Local VAEs by using the DCI scores (Eastwood & Williams, 2018). The DCI scores quantify the learned representations based on three types of aspects: Disentanglement, Compactness, and Informativeness. All metrics can be computed from the importance of each dimension of the latent space for predicting a factor of variation. DCI scores require the label information of the ground truth. Since the Local VAE clearly extracted the color information as the global feature, we calculated DCI scores for the two types of conditions: the class labels including all six aspects (w/ Color condition), and the class labels excluding color information (w/o Color condition).

Table 1 shows the empirical evaluations of the DCI scores. DCI scores with six labels (w/ Color condition) of the Local VAE were slightly better than the one for the Vanilla VAE. The model with small $\alpha$, which is the value closer to the Vanilla VAE, tended to achieve better scores for all DCI metrics in the Local VAE comparison. This result is attributed to the loss of color information from the internal representation as $\alpha$ increases. On the other hand, the Local VAEs significantly improved the DCI scores in the condition without color. All DCI metrics took their maximum value at $\alpha = 1$. The performance was slightly degraded at $\alpha = 1e1$, and the loss diverged during training at $\alpha = 1e2$. The numerical evaluation suggests that $\alpha$ can control how much of the structure behind the entire dataset is considered as global variation and from where it is regarded as a local variation. We also evaluated the performance of the $\beta$-VAE (Higgins et al., 2017) as a reference. The $\beta$-VAE modifies the KL term (Equation 1) by multiplying non-zero coefficient $\beta$. Although the $\beta$-VAE with $\beta = 8$ or $\beta = 16$ achieved higher scores than the Local VAE in the condition with color, the Local VAE with $\alpha = 1$ significantly outperformed all the $\beta$-VAE in the condition without color.

We also evaluated the quality of the generated images with the Fréchet Inception Distance (FID) (Heusel et al., 2017). FID is a metric that evaluates the similarity of quality between real and generated images. We used the 50,000 samples of the ground truth dataset and generated images for FID calculation. According to Table 1, FID tended to be low at the large $\alpha$ and took the minimum value at $\alpha = 1$. This result was consistent with the DCI scores of the condition without color. Since the method to calculate neighborhood is arbitrary, we also compared the synthetic neighborhood with a widely used approach. Please see Appendix D for the detail.

Table 1: Quantitative evaluations of the Local VAE on the 3D Shapes dataset. Highlighted cells indicate the model with the highest performance in the comparison of Local VAEs. Bold numbers indicate absolute best results.

| | | DCI w/ Color | | | DCI w/o Color | | | FID |
|---|---|---|---|---|---|---|---|---|
| | | Disent. | Compl. | Inform. | Disent. | Compl. | Inform. | |
| Local VAE | $\alpha = 0$ (Vanilla) | 0.246 | 0.204 | 0.703 | 0.150 | 0.096 | 0.547 | 134.786 |
| | $\alpha = 1e{-}3$ | 0.491 | 0.407 | 0.814 | 0.390 | 0.305 | 0.686 | 107.636 |
| | $\alpha = 1e{-}2$ | 0.449 | 0.385 | 0.797 | 0.173 | 0.132 | 0.635 | 123.288 |
| | $\alpha = 1e{-}1$ | 0.457 | 0.432 | 0.626 | 0.945 | 0.796 | 0.996 | 49.364 |
| | $\alpha = 1$ | 0.424 | 0.406 | 0.594 | **0.977** | **0.800** | **0.999** | **43.194** |
| | $\alpha = 1e1$ | 0.393 | 0.370 | 0.587 | 0.871 | 0.733 | 0.998 | 59.555 |
| $\beta$-VAE | $\beta = 2$ | 0.367 | 0.292 | 0.776 | 0.222 | 0.215 | 0.630 | 96.279 |
| | $\beta = 4$ | 0.588 | 0.499 | 0.906 | 0.384 | 0.337 | 0.817 | 96.612 |
| | $\beta = 8$ | 0.636 | **0.584** | **0.967** | 0.601 | 0.547 | 0.936 | 86.856 |
| | $\beta = 16$ | **0.649** | 0.580 | 0.941 | 0.690 | 0.473 | 0.883 | 86.237 |

## 5.2 CONCATENATED DATASET OF THE CARS3D AND SMALLNORB

Finally, we evaluated our proposed model on the dataset, which explicitly has the locality. In this section, we concatenated two datasets: the Cars3D Dataset (Reed et al., 2014) and the SmallNORB Dataset (LeCun et al., 2004). These datasets are both set of images of 3D objects. Each dataset has three (elevation, azimuth, and object type) and four (elevation, azimuth, category, and lighting condition) disentanglement factors, respectively. The elevation and azimuth are the global control factors common to the entire dataset, and the others are the sub-dataset specific factors. Each image has no information about which sub-dataset it comes from. We refer to this dataset as the CarsNORB Dataset in the following. Note that learning this entire dataset is more challenging than learning each sub-dataset respectively since these two sub-datasets have different structures.

Table 2 shows the empirical evaluation on the CarsNORB Dataset. The hyperparameters follow the same setting as the experiment on the 3D Shapes Dataset. We calculated the DCI Disentanglement score for each sub-dataset. According to the table, the Disentanglement score of the Vanilla VAE was remarkably low for the Cars3D Dataset. We believe this result comes from the difference in statistics that the Cars3D Dataset ($N = 17,568$) has fewer samples than the SmallNORB Dataset ($N = 48,600$) and has a more complicated structure, including colors. The Disentanglement scores took maximum at $\alpha = 1e{-}2$ for both sub-datasets. This result indicates that the locality enables the model to achieve the disentangled representation for each subspace without any label information.

Table 2: Quantitative evaluations of the Local VAE on the CarsNORB Dataset.

| | $\alpha = 0$ (Vanilla) | $\alpha = 1e{-}2$ | $\alpha = 1e{-}1$ | $\alpha = 1$ |
|---|---|---|---|---|
| NORB Disentanglement | 0.265 | **0.282** | 0.264 | 0.255 |
| Cars Disentanglement | 0.079 | **0.165** | 0.111 | 0.080 |

## 6 CONCLUSION

In this study, we proposed the Local VAE, a deep generative model suitable for datasets with local structure. Since conventional local learning approaches learn the embeddings at each neighborhood from scratch, integrating these approaches with deep neural networks, which require a massive amount of data and extended training time, was not reasonable. To overcome this trade-off, we performed gradient-based meta-learning, called MAML, with the supervision of past experiences outside the neighborhood. We evaluated our proposed model with the 3D Shapes dataset and the the concatenated dataset of the Cars3D and SmallNORB, which are one of the most straightforward datasets comprising disentangled local structures. Our experimental results showed that the learned representations of the Local VAE were more disentangled than that of the Vanilla VAE in terms of DCI scores. Moreover, the Local VAE improved the quality of the generated images compared with the Vanilla VAE according to subjective evaluation and FID scores.

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

# A    RELATIONSHIP BETWEEN THE LOCAL VAE AND A CONVENTIONAL LOCAL LEARNING

In this section, we discuss that the Local VAE can be interpreted as the variant of the conventional local learning approaches. We introduce Local Linear Embedding (LLE) (Roweis & Saul, 2000) as a typical local learning algorithm. The LLE extracts low-dimensional neighborhood-preserving embeddings based on the precomputed neighbor graph. This method assumes that the dataset consists of a combination of locally linear spaces, and applies a linear projection to each neighborhood. For the dataset $\mathcal{D} = \{\boldsymbol{x}^{(i)}\}_{i=1}^{N}$, the objective function of LLE is defined as

$$\mathcal{L}(W) = \sum_i \left\| \boldsymbol{x}^{(i)} - \sum_j W_{ij}\boldsymbol{x}^{(j)} \right\|^2, \tag{9}$$

where parameter $W$ is an $N \times N$ matrix. The element $W_{ij}$ of $W$ is nonzero only when $\boldsymbol{x}^{(j)}$ belongs to the set of neighbors of $\boldsymbol{x}^{(i)}$, and $\sum_j W_{ij} = 0$. The neighbor graph of $\boldsymbol{x}^{(i)}$ is built by using the $k$-nearest neighbor method. Since Equation 9 is known not to have local minima, we can derive the solution of Equation 9 by basic matrix calculation. Once the model parameter $W$ is derived, we can obtain the low-dimensional embeddings $\boldsymbol{z}^{(1)}, \boldsymbol{z}^{(2)}, \dots, \boldsymbol{z}^{(N)}$ of each data by minimizing the loss $\sum_i \|\boldsymbol{z}^{(i)} - \sum_j W_{ij}\boldsymbol{z}^{(j)}\|^2$ with respect to $\boldsymbol{z}$.

Here, we consider extending the embedding model of LLE from a linear projection to a general nonlinear model. The embedding model of $\boldsymbol{x}^{(i)}$ corresponds to $\sum_j W_{ij}\boldsymbol{x}^{(j)}$ in Equation 9. In other words, if we denote the index of the $\boldsymbol{x}^{(i)}$'s neighborhood as $j_1, \dots, j_K$, the model parameters of the neighborhood are $[W_{ij_1}, W_{ij_2}, \dots, W_{ij_K}]$. In the following, we generalize these $[W_{ij_1}, W_{ij_2}, \dots, W_{ij_K}]$ as parameter $\boldsymbol{\theta}_{N(\boldsymbol{x}^{(i)})}$ for the set of neighborhoods $N(\boldsymbol{x}^{(i)})$. Then, the aforementioned objective function is given by the following:

$$\mathcal{L}\left(\boldsymbol{\theta}_{N(\boldsymbol{x}^{(1)})}, \dots, \boldsymbol{\theta}_{N(\boldsymbol{x}^{(N)})}\right) = \sum_i \left\| \boldsymbol{x}^{(i)} - g_{\boldsymbol{\theta}_{N(\boldsymbol{x}^{(i)})}}\left(N(\boldsymbol{x}^{(i)})\right) \right\|^2. \tag{10}$$

Unlike Equation 9, there are no restrictions on the number of parameters or formulation, so optimization of the above equation is generally challenging. Notably, in the case of $g_{\boldsymbol{\theta}_{N(\boldsymbol{x}^{(i)})}}(\cdot)$ being a deep neural network, a massive amount of data and extended training time are required **for each $i$-th neighborhood** $N(\boldsymbol{x}^{(i)})$.

We can interpret Equation 10 as the general formulation of the Local VAE with the Gaussian Decoder. Consider taking only $\boldsymbol{x}^{(i)}$ itself instead of a set of neighborhoods $N(\boldsymbol{x}^{(i)})$ of $\boldsymbol{x}^{(i)}$ as input to the function $g_{\boldsymbol{\theta}_{N(\boldsymbol{x}^{(i)})}}(\cdot)$ in Equation 10. If we take the model parameters as $\boldsymbol{\theta}_{N(\boldsymbol{x}^{(i)})}$ and $\boldsymbol{\phi}_{N(\boldsymbol{x}^{(i)})}$ and use Autoencoder for the model $g(\cdot)$, Equation 10 corresponds to the objective function of the Local VAE with the Gaussian Decoder.

# B    EXPERIMENTAL CONDITIONS AND HYPERPARAMETERS

In this section, we show the experimental conditions and hyperparameters which are used for all the numerical experiments in the main text. Table 3 shows the Encoder and the Decoder architectures of the VAE. We used the multivariate isotropic Gaussian for the latent variable. The outputs of the Encoder correspond to $\boldsymbol{\mu}$ and $\log \boldsymbol{\sigma}$ of the variational posterior distribution $q(\boldsymbol{z}|\boldsymbol{x})$. Table 4 shows the hyperparameters for the model and the training procedure. In addition to the parameters shown in the table, we used the gradient boosted trees from Scikit-learn with the default setting for computing the DCI scores. We also used the Inception-v3 network from Keras, which is pre-trained on the ImageNet dataset to compute the FID.

Table 3: Network architecture for the numerical experiments.

| Encoder | Decoder |
|---|---|
| Input: $64 \times 64 \times 3$ | Input: $\mathbb{R}^{10}$ |
| $4 \times 4$ conv, 32 ReLU, stride 2 | FC, 256 ReLU |
| $4 \times 4$ conv, 32 ReLU, stride 2 | FC, $4 \times 4 \times 64$ ReLU |
| $4 \times 4$ conv, 64 ReLU, stride 2 | $4 \times 4$ upconv, 64 ReLU, stride 2 |
| $4 \times 4$ conv, 64 ReLU, stride 2 | $4 \times 4$ upconv, 32 ReLU, stride 2 |
| FC 256, F2 $2 \times 10$ | $4 \times 4$ upconv, 32 ReLU, stride 2 |
| | $4 \times 4$ upconv, 3, stride 2 |

Table 4: The hyperparameters which are used for the numerical experiments.

| Parameter | Value |
|---|---|
| Batch size (corresponds to the number of tasks) | 25 |
| Inner batch size (corresponds to $K$) | 10 |
| Latent space dimension | 10 |
| Optimizer | Adam |
| Adam: beta1 | 0.9 |
| Adam: beta2 | 0.999 |
| Adam: epsilon | 1e−8 |
| Adam: learning rate | 1e−4 |
| Decoder type | Bernoulli |
| Training steps | 300,000 |

## C    LATENT INTERPOLATION

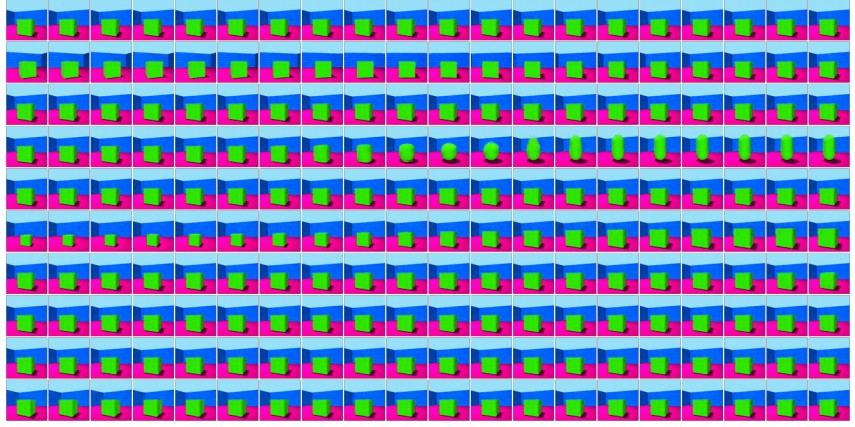

Figure 4: Interpolation of Local VAE's latent space. Each $i$-th row corresponds to the reconstructed image with the latent code $z_i$ modified in the range of $[-2, 2]$.

Figure 4 shows the learned latent space of the Local VAE model. We swept each latent dimension for the specific training sample in the range of $[-2, 2]$. The model extracted the angle, shape, and size of the object as the disentangled factors. The color of the reconstructed images was not changed against the latent space interpolation. We believe that this is because the model extracted the color information as a global feature.

## D    COMPARISON OF $k$-NEIGHBOR CONSTRUCTION METHODS

As mentioned in the main text, we can arbitrarily choose the distance for neighborhood construction in Local VAE. We evaluated two types of methods: synthetic neighborhood by sampling and k-

nearest neighborhood on latent space. Among them, although the synthetic neighborhood has the advantage that it is simple to implement and fast to compute, $\ell^2$ distance on input space is widely used in general. In this section, we compare the performance of the synthetic neighborhood to the $\ell^2$ distance on input space.

Table 5 shows the comparison of the methods to compute neighborhood. We used the 3D Shapes Dataset and set the hyperparameter as $\alpha = 1$. Both methods outperformed Vanilla VAE, and they achieved comparable scores with each other. We note that the synthetic neighborhood by sampling achieved slightly better performance than the $\ell^2$ distance on input space in this experiment. This result suggests that there is an appropriate distance for each dataset, which is not necessarily the $\ell^2$ distance on input space. Choosing the appropriate distance will improve the quality of the learned representations and generated images in practical use.

Table 5: Quantitative comparison of the neighborhood construction on the 3D Shapes dataset.

| | DCI w/ Color | | | DCI w/o Color | | | FID |
|---|---|---|---|---|---|---|---|
| | Disent. | Compl. | Inform. | Disent. | Compl. | Inform. | |
| Sampling | 0.424 | 0.406 | 0.594 | 0.977 | 0.800 | 0.999 | 43.194 |
| $\ell^2$ on input space | 0.397 | 0.342 | 0.582 | 0.676 | 0.520 | 0.813 | 49.187 |
| Vanilla VAE | 0.246 | 0.204 | 0.703 | 0.150 | 0.096 | 0.547 | 134.786 |

