# OpenReview forum: "Localized Generations with Deep Neural Networks for Multi-Scale Structured Datasets"
_ICLR.cc/2020/Conference — Reject_

### Official Review · AnonReviewer3 · 2019-10-19
**Official Blind Review #3**

**Rating:** 3

**Review:**

This paper proposed a local VAE based on the model-agnostic meta-learning concept. Images generated based local VAE are shown to be better than those generated bu \beta-VAE in general in terms of generation quality and disentanglement/compactness/informativeness.

This paper shows an interesting idea of viewing neighbourhood of a data point as a task to adopt the meta-learning concept to  train VAE which gives superior performance.

However, this paper falls short regarding its overall organisation of the paper and the way that different concepts being articulated which make it hard to follow and read. It is not easy to understand the linkage between a task and a neighbourhood being advocated without re-reading the paper for a few times. It seems to me that very substantial effort is needed for the revision to reach the ICLR standard.

The introduction section should be carefully rewritten to explain clearly the notions of locality and structural relationship. As far as I can see in the subsequent sections, the locality/structural relationship seems to be referring to the neighbourhood either in the input space or the latent space. Also, the way of discussing local learning should be elaborated a bit to make it clear what exact notion of local learning  is being presented.


Specific comments:

Page 2:
“since the most dataset tends to be governed by the consistent rules of the physical world”
->
“since most dataset tends to be governed by the consistent rules of the physical world”

“Such kind of dataset has a multi-scale structure from a local to a global scale.” - some grammatical issue

Page 5:
Line 4: Equation 3 seems not related to meta-learning


**Experience Assessment:**

I have published one or two papers in this area.

**Review Assessment: Checking Correctness Of Derivations And Theory:**

I assessed the sensibility of the derivations and theory.

**Review Assessment: Checking Correctness Of Experiments:**

I assessed the sensibility of the experiments.

**Review Assessment: Thoroughness In Paper Reading:**

I read the paper at least twice and used my best judgement in assessing the paper.

---

> ### Author Response · Authors · 2019-11-11
> **Answer to Reviewer #3**
>
> Thank you for carefully read our manuscript. We would like to address your concerns about the organization of the paper below.
>
> > However, this paper falls short regarding its overall organisation of the paper and the way that different concepts being articulated which make it hard to follow and read. It is not easy to understand the linkage between a task and a neighbourhood being advocated without re-reading the paper for a few times.
>
> Although the idea of ​​executing MAML by considering each neighborhood as a task is an original part of this study, there was not enough explanation, as you pointed out. We will reconstruct the introduction to make this part much more clear.
>
> Moreover, we introduced LLE in Sect. 2.1 to describe the interpretation that our Local VAE with Gaussian decoder can be regarded as an extension of the LLE as we discussed in p.4, last paragraph. It will help to find how our proposed model relates to the conventional local learning approach. On the other hand, multiple concepts make the manuscript hard to follow. We would like to move the Sect. 2.1 before the Sect. 4 or to the Appendix.
>
> > The introduction section should be carefully rewritten to explain clearly the notions of locality and structural relationship. As far as I can see in the subsequent sections, the locality/structural relationship seems to be referring to the neighbourhood either in the input space or the latent space. Also, the way of discussing local learning should be elaborated a bit to make it clear what exact notion of local learning  is being presented.
>
> We assumed the dataset that we are interested in often has the local structure which changes on location in observation space. We referred to this property as “locality” and also assumed these local structures are similar to each other. This assumption corresponds to the “structural similarity.” These assumptions mean that two-scale structures appear in the dataset: the local structure inside each neighborhood and the global relationship between each neighborhood. Figure 1 in our manuscript is the schematic diagram visualizing these two scales.
>
> However, as you pointed out, the description in the manuscript seems to be a little bit confusing. We will organize the usage of "locality" and "structural relationship" and clearly state the relation to the “neighborhood.” This study proposed the learning algorithm for the dataset with a local structure on some distance. We described the actual realization of this distance in Sect. 3.1 because it is independent of the main claim. We will clearly state this hierarchy of concepts in the introduction.
>
> > Page 5, Line 4: Equation 3 seems not related to meta-learning
>
> The MAML minimizes the normal objective function in the inner-loop. In the Local VAE, eq. (3) corresponds to the normal objective function.

---

### Official Review · AnonReviewer1 · 2019-10-23
**Official Blind Review #1**

**Rating:** 3

**Review:**

Authors of this paper propose to utilize the embedding model for the other local structure as a weak form of supervision based on the insight that the real-world datasets often shares some structural similarity between each neighborhood. The Local VAE is proposed to have the different model parameters for each local subset and train these local parameters by the gradient-based meta-learning.

Local VAE incorporates local information by using prior distributions of local parameters in VAE. The overall model performs probabilistic inference via the conditional distribution from the meta parameters. There are several concerns:
1. In section 2, authors discussed LLE. It is unclear the purpose of the section 2.1?
2. LLE does not require W is nonnegative only, and \sum_j W_{i,j}=0 is also contradictory with the nonnegative assumption.
3. Authors claimed that Local VAE algorithm corresponds to the assumption that the dataset approximately lies on multiple subsets and each subset is generated from different parameters. It is unclear what is the connection of the Local VAE to multi-scale structures of the datasets.
4.  Authors evaluated neighbors by sampling and k-nearest neighbors on latent space. It is unclear why not use the common k-nearest neighbors on the input data. K-nearest search should not be a computational problem for large datasets by using fast approach.

As the motivations of this paper, existing methods require massive amount of data and extended training time. However, authors did not demonstrate these points by comparing the proposed method with existing methods.


**Experience Assessment:**

I have read many papers in this area.

**Review Assessment: Checking Correctness Of Derivations And Theory:**

I assessed the sensibility of the derivations and theory.

**Review Assessment: Checking Correctness Of Experiments:**

I assessed the sensibility of the experiments.

**Review Assessment: Thoroughness In Paper Reading:**

I read the paper at least twice and used my best judgement in assessing the paper.

---

> ### Author Response · Authors · 2019-11-11
> **Answer to Reviewer #1**
>
> Thank you for your insightful discussions and comments. We would like to address your concerns one by one. Please see below for the details.
>
> > 1. In section 2, authors discussed LLE. It is unclear the purpose of the section 2.1?
>
> We introduced LLE in Sect. 2.1 to describe the interpretation that our Local VAE with Gaussian decoder can be regarded as an extension of the LLE as we discussed in p.4, last paragraph. It will help to find how our proposed model relates to the conventional local learning approach. On the other hand, as R3 and you pointed out, the multiple concepts make the manuscript hard to follow. We would like to move the Sect. 2.1 before the Sect. 4 or to the Appendix.
>
> > 2. LLE does not require W is nonnegative only, and $\sum_j W_{i,j}=0$ is also contradictory with the nonnegative assumption.
>
> We intended to write as "The element W_ij of W is nonzero only when x^j belongs to ..." but accidentally wrote as "nonnegative." Thank you for pointed out. We will replace the text.
>
> > 3. Authors claimed that Local VAE algorithm corresponds to the assumption that the dataset approximately lies on multiple subsets and each subset is generated from different parameters. It is unclear what is the connection of the Local VAE to multi-scale structures of the datasets.
>
> The main claim of our manuscript is that when the dataset has the locality and each neighborhood shares structural similarity, we can efficiently (in the sense of the size of the dataset and the training time) train the deep neural networks in a local learning manner. It is reasonable to assume that such a dataset has two scales of structures: the local structure inside each neighborhood and the global relationship between each neighborhood. Figure 1 in our manuscript is the schematic diagram visualizing these two scales. Based on this context, we called the dataset as "multi-scale structured." However, this phrase itself is a little bit vague; we might be able to recall another meaning. We will replace the phrase "multi-scale structure" to "locality and/or structural similarity" because the phrase itself is not essential to our claim.
>
> > 4. Authors evaluated neighbors by sampling and k-nearest neighbors on latent space. It is unclear why not use the common k-nearest neighbors on the input data. K-nearest search should not be a computational problem for large datasets by using fast approach.
>
> In the experiment of the Shapes3D dataset, we employed the synthetic neighbors by sampling because it was much faster and simpler to implement than the original k-neighbor on the input space. The main claim of our study is independent of the actual realization of the neighborhood, and numerical experiments show that our model already obtained superior performance compared to methods that do not use the local information (the Vanilla VAE and beta-VAE). Based on this fact, we believe that we showed the effectiveness of our approach, even the neighborhood construction is synthetic. Of course, we can expect further improvement in performance by adopting a more appropriate neighborhood.
>
> Also, in the experiment of the CarsNORB dataset, we employed k-neighbors in the latent space because the latent space of the NN will have a more "natural" (which close to our intuition) distance than the input space [e.g. Caron+, ICCV 2019].
>
> Since there is no reason not to use the input space actively, it is possible to add a performance comparison of the method to calculate the neighborhood.
>
> > 5. As the motivations of this paper, existing methods require massive amount of data and extended training time. However, authors did not demonstrate these points by comparing the proposed method with existing methods.
>
> We did not demonstrate the efficiency of our proposed model, because we think that it is obvious by the construction of the method. As we mentioned in Sect. 2.1, when we apply the conventional local learning approach to the nonlinear embedding model, we need to train the embedding model from scratch for each i-th neighborhood N(x^i). For example, in the case of the Shapes3D dataset, the number of data points is 480,000, which means that the conventional approach needs to train neural networks with different parameters 480,000 times. Furthermore, because local learning uses only a little data for each neighborhood in general, it is hard to train a deep neural network in such a situation. Therefore, it is practically impossible to train the model until it converges for each neighborhood of all data.

---

### Official Review · AnonReviewer2 · 2019-10-23
**Official Blind Review #2**

**Rating:** 8

**Review:**

Incorporating the local learning approaches into the training of the deep generative models can potentially create a new model that has both the capacity for high-dimensional inputs and flexibility for locally changing environments. However, the local learning approach is limited to using only a simple model, because complex models require a large amount of data and extended training time. This paper overcomes this trade-off based on the insight that the real-world dataset often shares some structural similarities between each neighborhood.

Pros:
The paper is well-written. It is easy for the reader to understand. The derivations in the paper are correct.


The motivation is plausible. It is reasonable to expect that each local subspace of the dataset shares some structure since most dataset tends to be governed by the consistent rules of the real world.


The numerical experiments show that the locality enables the model to achieve the disentangled representation for each subspace without any label information.

Cons:
The novelty seems a little straight-forward. The paper just extends the objective function of the VAE to have different parameters for each local subset.

In consideration of extended training time in the complex model, the paper doesn't provide an evaluation of the efficiency of their proposed model.

The paper isn't very polished yet. There were more than a few spelling and grammatical errors, please proofread the work and improve the writing.

This paper in its current form is already fairly good.

**Experience Assessment:**

I do not know much about this area.

**Review Assessment: Checking Correctness Of Derivations And Theory:**

I assessed the sensibility of the derivations and theory.

**Review Assessment: Checking Correctness Of Experiments:**

I assessed the sensibility of the experiments.

**Review Assessment: Thoroughness In Paper Reading:**

I made a quick assessment of this paper.

---

> ### Author Response · Authors · 2019-11-11
> **Answer to Reviewer #2**
>
> Thank you for the review and comments for the paper.
>
> We are glad that you assessed the organization, derivation, and experiments of our paper, and evaluated the paper in its current form as already fairly good. But, we will reconstruct the paper since there are still some rooms to improve as R1 and R3 mentioned.
>
> Next, we discuss your concerns regarding the efficiency of our model. R1 made a similar point regarding your concern “In consideration of extended training time in the complex model, the paper doesn't provide an evaluation of the efficiency of their proposed model.” We would like to state the same reply to R1 below:
>
> We did not demonstrate the efficiency of our proposed model, because we think that it is obvious by the construction of the method. As we mentioned in Sect. 2.1, when we apply the conventional local learning approach to the nonlinear embedding model, we need to train the embedding model from scratch for each i-th neighborhood N(x^i). For example, in the case of the Shapes3D dataset, the number of data points is 480,000, which means that the conventional approach needs to train neural networks with different parameters 480,000 times. Furthermore, because local learning uses only a little data for each neighborhood in general, it is hard to train a deep neural network in such a situation. Therefore, it is practically impossible to train the model until it converges for each neighborhood of all data.

---

### Author Response · Authors · 2019-11-12
**Summary of revision**

Dear reviewers,

Thank you for your detailed review and constructive feedback. Especially, R1 and R2 gave us helpful suggestions for the structure of the manuscript. We updated our manuscript based on your comments and discussions. We restructured mainly the organization of the introduction section, to make it easy to follow. The summary of the changes is shown below.

R1 pointed out that the connection of the Local VAE to multi-scale structures of the datasets is unclear. The phrase "multi-scale structure" itself was ambiguous a little, so we rephrased it with locality and/or structural similarity.

R1 pointed out that the purpose of section 2.1 that introduces LLE is unclear, and R3 stated that the paper in the current form is hard to follow because "the way that different concepts being articulated." Based on these suggestions, we moved the section about LLE to the appendix.

R3 pointed out that it is not easy to understand the linkage between a task and a neighborhood. R3 also asked us to explain clearly the notions of locality and structural relationship. We reconstructed the paragraph about structural similarity and meta-learning. We clearly stated our assumption about the dataset and the relationship between a task and a neighborhood (introduction, paragraphs 4 and 5). We also added a brief explanation about the local learning in the introduction, paragraph 2.

---

> ### Author Response · Authors · 2019-11-14
> **Additional experiment on the comparison of neighborhood construction**
>
> To address R1's concern, we conducted additional experiments on neighborhood construction. In appendix D, we compared the performance of the l2 distance on input space and the synthetic neighborhood by sampling. We only conducted the experiment on a specific hyperparameter due to the time limit.

---

### Decision · Program_Chairs · 2019-12-19

**Decision:**

Reject

**Comment:**

The paper presents a structured VAE, where the model parameters depend on a local structure (such as distance in feature or local space), and it uses the meta-learning framework to adjust the dependency of the model parameters to the local neighborhood.

The idea is natural, as pointed by Rev#1. It incurs an extra learning cost, as noted by Rev#1 and #2, asking for details about the extra-cost. The authors' reply is (last alinea in first reply to Rev#1): we did not comment (...) because in essence, using neighborhoods in a naive way is not affordable.
The area chair would like to know the actual computational time of Local VAE compared to that of the baselines.

More details (for instance visualization) about the results on Cars3D and NORB would also be needed to better appreciate the impact of the locality structure. The fact that the optimal value (wrt Disentanglement) is rather low ($10^{-2}$) would need be discussed, and assessed w.r.t. the standard deviation.

In summary, the paper presents a good idea. More details about its impacts on the VAE quality, and its computation costs, are needed to fully appreciate its merits.